# Modulation of miR-29a and ADAM12 Reduces Post-Ischemic Skeletal Muscle Injury and Improves Perfusion Recovery and Skeletal Muscle Function in a Mouse Model of Type 2 Diabetes and Peripheral Artery Disease

**DOI:** 10.3390/ijms23010429

**Published:** 2021-12-31

**Authors:** Victor Lamin, Joseph Verry, Isaac Eigner-Bybee, Jordan D. Fuqua, Thomas Wong, Vitor A. Lira, Ayotunde O. Dokun

**Affiliations:** 1Division of Endocrinology and Metabolism, Carver College of Medicine, University of Iowa, Iowa City, IA 52242, USA; victor-lamin@uiowa.edu (V.L.); joseph-verry@uiowa.edu (J.V.); isaac-eigner-bybee@uiowa.edu (I.E.-B.); thomas-wong@uiowa.edu (T.W.); 2Department of Health and Human Physiology, College of Liberal Arts and Sciences, University of Iowa, Iowa City, IA 52242, USA; jordan-fuqua@omrf.org (J.D.F.); vitor-lira@uiowa.edu (V.A.L.); 3Fraternal Order of Eagles Diabetes Research Center, Carver College of Medicine, University of Iowa, Iowa City, IA 52242, USA

**Keywords:** miR-29a, ADAM12, diabetes mellitus, high fat diet and PAD

## Abstract

Both Type 1 diabetes mellitus (DM1) and type 2 diabetes mellitus (DM2) are associated with an increased risk of limb amputation in peripheral arterial disease (PAD). How diabetes contributes to poor PAD outcomes is poorly understood but may occur through different mechanisms in DM1 and DM2. Previously, we identified a disintegrin and metalloproteinase gene 12 (ADAM12) as a key genetic modifier of post-ischemic perfusion recovery. In an experimental PAD, we showed that ADAM12 is regulated by miR-29a and this regulation is impaired in ischemic endothelial cells in DM1, contributing to poor perfusion recovery. Here we investigated whether miR-29a regulation of ADAM12 is altered in experimental PAD in the setting of DM2. We also explored whether modulation of miR-29a and ADAM12 expression can improve perfusion recovery and limb function in mice with DM2. Our result showed that in the ischemic limb of mice with DM2, miR-29a expression is poorly downregulated and ADAM12 upregulation is impaired. Inhibition of miR-29a and overexpression of ADAM12 improved perfusion recovery, reduced skeletal muscle injury, improved muscle function, and increased cleaved Tie 2 and AKT phosphorylation. Thus, inhibition of miR-29a and or augmentation of ADAM12 improves experimental PAD outcomes in DM2 likely through modulation of Tie 2 and AKT signalling.

## 1. Introduction

Peripheral artery disease (PAD) is an atherosclerotic, occlusive disease of the lower extremities, which may result in serious tissue damage and death [1]. Approximately 50% of patients with PAD are asymptomatic, making it difficult to estimate its true prevalence [1,2]. However, available data estimated that worldwide more than 200 million people have PAD, with a spectrum of symptoms ranging between claudication, ischemic rest pain, and ischemic ulcerations [1,2]. Diabetes mellitus (DM) is one of the major risk factors for developing PAD [3,4,5,6]. When DM is present in individuals with PAD, the outcomes are worse with an increased likelihood of death [3,4,5,6]. Furthermore, diabetic patients with PAD are more likely to have disease in arteries distal to the knee and are five times more likely to have disease requiring an amputation than non-diabetic patients with PAD [7,8]. DM can modify PAD outcomes by accelerating the progression of atherosclerotic occlusive disease, but it can also modify post-ischemic adaptive mechanisms involved in restoring blood flow following vessel occlusion [6,8,9,10,11].

The clinical management of PAD has evolved over the past decade to include a broad approach, focusing on the reduction of adverse cardiovascular events, improving symptoms in claudication, and preventing tissue loss in critical limb ischemia [12]. However, current medical therapies were derived from studies designed to treat coronary artery disease and prevent acute thrombotic occlusion [13,14]. There are no medical therapies available that have shown the ability to significantly improve leg blood flow in patients with PAD and thus no medical therapies are available to directly treat the primary problem of reduced blood flow [15,16,17,18]. Therefore, new treatment paradigms are needed for the effective clinical management of PAD.

Gene therapy and cell-based therapy have emerged as novel approaches to promote angiogenesis in order to improve leg blood flow in patients with PAD [19,20,21,22,23]. An increase in gene expression can be achieved by either ectopic expression of the gene or by manipulating its regulatory mechanisms. microRNAs (miRNAs) are now recognized as one of the key post transcriptional regulators of many genes [24]. miRNAs are a class of small, non-coding, single-stranded RNAs, 19–23 nucleotides in length, which target the 3′UTR, 5′ UTR, or coding region of mRNAs, resulting in mRNA degradation or repression of translation into protein [24].

Using an experimental PAD or hind limb ischemia (HLI) model in mice, we previously identified a genetic locus in mice termed the Limb Salvage QTL 1 (LSQ-1) that is associated with favorable post-ischemic perfusion recovery and limb preservation [25]. Within LSq-1 we identified a disintegrin and metalloproteinase gene 12 (ADAM12) as a gene that mediates some of the key effects associated with this QTL [25]. Additionally, we showed that ADAM12 is upregulated in ischemic mouse GA muscle and this upregulation is impaired in mice with DM1 and associated with poor post-ischemic perfusion recovery [25,26]. Overexpression of ADAM12 in ischemic GA muscle of mice with type 1 DM improved perfusion recovery. Thus, the expression of ADAM12 in ischemic limbs of mice with DM could potentially be a new treatment paradigm to improve blood flow in PAD. However, given post-ischemic limbs perfusion recovery is impaired in mice with type 2 DM, it is possible that ADAM12 upregulation is also impaired in ischemic DM2 limbs. Moreover, whether overexpression of ADAM12 in ischemic GA muscle of mice with DM2 would improve perfusion recovery following experimental PAD is not known.

Previously, we showed that in ischemic mouse GA muscle, miR-29a expression is downregulated and allows upregulation of ADMA12 [25]. In mice with DM1, this process is impaired such that miR-29a expression is not downregulated, resulting in impaired ADAM12 upregulation [26]. Whether there is impaired miR-29a regulation in ischemic GA muscle of mice with DM2 is not known. Moreover, whether the modulation of miR-29a expression could improve post-ischemic perfusion recovery in mice with DM2 has not been shown. Lastly, it is not known whether the modulation of gene expression through manipulation of microRNA is as effective as overexpression of its target gene through direct cDNA transfer in ischemic DM tissues.

In this study, we assessed whether there is an altered expression of ADAM12 and miR-29a in ischemic GA muscle of high fat fed (HFD) mice as a model of DM2. We also tested whether the overexpression of ADAM12 or inhibition of miR-29a expression could improve post-ischemic limbs perfusion recovery in DM2 mice. Moreover, given that one of the goals of therapy in humans with PAD is improving limb function and reducing limb amputation as a result of ischemic injury, we explored whether these treatments would reduce skeletal muscle injury and improve muscle function in the treated mice. Lastly, we explored possible physiologic and molecular mechanisms by which ADAM12 gene transfer and miR-29a inhibition may be improving outcomes following experimental PAD in DM2 mice.

## 2. Results

### 2.1. HFD Feeding Impairs Ischemia Induced ADAM12 Upregulation and miR-29a Downregulation in C57BL/6 Mice

We have previously shown that there is impaired regulation of miR-29a in the ischemic gastrocnemius muscle (GA) of Akita mice a model of type 1 DM [26]. Here we assessed miR-29a expression in the ischemic GA muscle of HFD mice, a model of type 2 diabetes. miR-10a was used as a loading control as it has been shown to be stable in ischemic and non-ischemic conditions (Appendix A). Our results show, unlike normal chow fed mice where miR-29a expression is downregulated in the ischemic GA muscle, in HFD mice, miR-29a remains elevated (Normal Chow ischemic: 0.31 ± 0.01 vs. HFD ischemic: 0.60 ± 0.13 * *p* < 0.05 and Figure 1A). In our prior study, we showed that ADAM12 is down-regulated by miR-29a. In this study, we assessed whether the elevated miR-29a expression in ischemic HFD mouse GA muscle is associated with impaired upregulation of ADAM12. We found impaired upregulation of ADAM12 in ischemic HFD mouse GA muscle (Normal Chow ischemic: 16.96 ± 2.72 vs. HFD ischemic: 4.04 ± 1.69, ** *p* < 0.005 and Figure 1B).

### 2.2. ADAM12 Gene Transfer and miR-29a Inhibition in Ischemic GA Muscle of HFD Mice Increased ADAM12 Expression, Improved Perfusion Recovery and Capillary Density

Given we observed impaired downregulation of miR-29a and its target gene ADAM12 in the ischemic GA muscle of HFD mice, we investigated the effect of plasmid-mediated overexpression of ADAM12 and miR-29a inhibition in the ischemic GA muscle of HFD mice. Our results show both ADAM12 gene transfer and miR-29a inhibition increased ADAM12 mRNA (normal chow: 1.0 ± 0.17, HFD: 0.08 ± 0.02, HFD + ADAM12: 0.73 ± 0.10 and HFD + miR-29aINH: 7.41 ± 0.61, ** *p* < 0.005. Figure 2A) and protein expression (normal chow: 1.0 ± 0.20, HFD: 0.11 ± 0.04, HFD + ADAM12: 0.41 ± 0.14 and HFD + miR-29aINH: 1.52 ± 0.53, and * *p* < 0.05. Figure 2B,C). 

It is well established that following experimental PAD induction in HFD mice, perfusion recovery is impaired [27,28]. Moreover, we have previously shown that ADAM12 plays a key role is perfusion recovery following experimental PAD in non-diabetic mice [25]. We therefore hypothesized that impaired upregulation of ADAM12 may play a role in impaired perfusion recovery in HFD mice. Hence, we explored the therapeutic effectiveness of ADAM12 augmentation. Moreover, we previously found impaired downregulation of miR-29a in ischemic type 1 DM mice contributed to poor post-ischemic perfusion recovery [26]. We therefore hypothesized that impaired miR-29a downregulation in post-ischemic GA muscle may contribute to poor perfusion recovery in type 2 DM. Here we assessed the effects of ADAM12 augmentation and miR-29a inhibition on perfusion recovery and capillary density in the ischemic GA muscle of HFD mice (a model of type 2 DM) following the induction of experimental PAD. Our results show both ADAM12 augmentation and miR-29a inhibition improved perfusion recovery in treated mice compared to controls (day-21 percent perfusion recovery: Normal chow: 84.47 ± 4.33, HFD: 42.52 ± 5.35, HFD + ADAM12: 58.45 ± 4.87, HFD + miR-29aINH: 97.59 ± 6.14, * *p* < 0.05, Figure 3A,B). We also performed immunohistochemistry for CD31^+^ cells to quantify capillary density in the ischemic GA muscle by counting the total number of CD31^+^divided by the total number of myofibers in the GA muscle, and found on day 21 post-HLI, ADAM12 gene transfer and miR-29a inhibition lead to a significant increase in the number of stained capillaries per muscle fibers (normal chow: 1.00 ± 0.03, HFD: 0.74 ± 0.02, HFD + ADAM12: 1.37 ± 0.07, and HFD + miR-29aINH: 1.78 ± 0.15, * *p* < 0.05, Figure 3C,D). 

Additionally, to assess tissue-specific delivery of the miR-29a inhibitor, we quantified the expression of miR-29a in various organs of treated mice at day 3 and week 3, and the result shows that there was no change in miR-29a expression levels, suggesting that our drug delivery method by ultrasound-induced microbubbles was able to target delivery primarily to the limb (Appendix A).

### 2.3. ADAM12 Augmentation and miR-29a Inhibition Reduced the Extent of Skeletal Muscle Injury and Increased Skeletal Muscle Function in Ischemic GA Muscle of HFD Mice

We next investigated whether ADAM12 augmentation or miR-29a inhibition could protect against ischemic injury following experimental PAD in HFD mice. Groups of mice were treated as described in the method. At day 21 following experimental PAD, ischemic GA muscles were harvested, and sections were stained with hematoxylin and Eosin. Centrally located nuclei are a known indication of skeletal muscle injury, hence we assessed the number of muscle fibers with centralized nuclei by counting the total number of centrally located nuclei divided by the total number of myofibers in the GA muscle [29]. Our results showed an increased percentage of muscle fibers with centrally located nuclei in muscle sections from HFD mice (normal chow: 1.4% ± 0.6, HFD: 25.2% ± 4.3). Sections from HFD mice treated with ADAM12 or the miR-29a inhibitor were found to have a significantly decreased percentage of muscle fibers with centrally located nuclei (HFD + ADAM12: 13.9% ± 4.1, and HFD + miR-29aINH: 4.0% ± 1.2, * *p* < 0.05, ** *p* < 0.005, Figure 4A,B). Additionally, we investigated if the maximal isometric torque, as a measure of muscle force production, was affected by these interventions in HFD mice. Our data also showed at day 21 following the induction of experimental PAD, HFD mice were weaker at the post-ischemic hind limb when compared to normal-chow-fed mice. Both the overexpression of ADAM12 and the inhibition of miR-29a improved skeletal muscle force production over non-treated HFD mice (normal chow: 0.24 ± 0.06, HFD: 0.11 ± 0.03, HFD + ADAM12: 0.26 ± 0.06 and HFD + miR-29aINH: 0.48 ± 0.09, * *p* < 0.05 and ** *p* < 0.005, Figure 4C).

### 2.4. ADAM12 Gene Augmentation and miR-29a Inhibition Increased Endothelial Progenitor Cells in Post Ischemic GA Muscle of HFD Mouse

Prior studies have shown the role of endothelial progenitor cells in mediating post-ischemic angiogenesis in experimental PAD [30,31]. There are also data to suggest the extent of recruitment of these progenitor cells is impaired in the setting of DM [32]. Our data shown above indicated ADAM12 gene transfer and miR-29a inhibition increased CD31^+^ endothelial cells per muscle fiber in our immunohistochemical studies (Figure 3C,D). To gain some insight into the physiologic mechanism by which ADAM12 gene transfer and miR-29a inhibition improved perfusion recovery in HFD mice, we hypothesized that this may be occurring through improved recruitment of endothelial progenitor cells. We therefore assessed the extent of endothelial progenitor cells recruited into the ischemic mouse GA muscle of HFD mice using flow cytometry. We quantified the percentage of infiltrating cells expressing CD45^−^CD31^+^CD34^+^, which are known endothelial progenitor cells makers [33]. Our results show that both ADAM12 gene transfer and treatment with miR-29a inhibitor led to an increase in the percentage of CD45^−^CD31^+^CD34^+^ cells as compared to the control untreated group (normal chow: 0.69% ± 0.08, HFD: 0.25% ± 0.03, HFD + ADAM12: 0.55% ± 0.08, and HFD + miR-29aINH: 0.58% ± 0.11. * *p* < 0.05, Figure 5A–F)

### 2.5. miR-29a Inhibition and ADAM12 Augmentation Modulate Tie2 Cleavage through Increased Akt Phosphorylation (P-AKT) In-Vivo

ADAM12 has been previously reported to be expressed by endothelial cells and can mediate Tie2 activation by ectodomain shedding/cleavage [34]. In a hind limb ischemic model of PAD, Tie2 was shown to play a critical role in perfusion recovery [35] and its activation resulted in phosphorylation of Akt (p-AKT) [36,37]. We hypothesized that both miR-29a inhibition and ADAM12 augmentation in vivo may modulate perfusion recovery, in part, through increased expression of cleaved Tie2. 

We assessed in vivo whether increased ADAM12 expression in the ischemic GA muscle was associated with increased Tie2 shedding. Our result shows increased levels of cleaved Tie2 following ADAM12 gene overexpression and miR-29a inhibition (Normal Chow: 1.0 ± 0.30, HFD Control: 0.31 ± 0.05, HFD + ADAM12: 1.93 ± 0.65, and HFD + miR-29aINH: 2.40 ± 0.65. HFD Control vs. HFD + ADAM12 * *p* < 0.05, ** *p* < 0.005 and Figure 6A,B and HFD Control vs. HFD + miR-29aINH, * *p* < 0.05, ** *p* < 0.005 and Figure 6A,B. 

Furthermore, we investigated the phosphorylation status of Akt (p-AKT) in the day 3 post-ischemic GA muscle of HFD mice treated with ADAM12 overexpression or miR-29a inhibition. We observed that ectopic overexpression of ADAM12 or miR-29a inhibition was associated with increased Akt phosphorylation. (Normal Chow: 1.00 ± 0.36, HFD Control: 0.24 ± 0.03, HFD + ADAM12: 0.75 ± 0.12, and HFD + miR-29aINH: 1.01 ± 0.16. HFD Control vs. HFD + ADAM12 * *p* < 0.05, ** *p* < 0.005 and Figure 6C,D and HFD Control vs. HFD + miR-29aINH, * *p* < 0.05, ** *p* < 0.005 and Figure 6C,D).

## 3. Discussion

Previously, we identified ADAM12 as a gene that played a key role in perfusion recovery following experimental PAD [25]. ADAM12 is upregulated in ischemic mouse GA and contributes to post-ischemic perfusion recovery but its upregulation is impaired in ischemic GA of mice with type 1 diabetes and associated with poor post-ischemic perfusion recovery [25,26]. Furthermore, we showed that impaired ADAM12 upregulation in ischemic GA of mice with type 1 diabetes was due to impaired downregulation of miR-29a, a miRNA that regulates ADAM12 expression in ischemia [25]. We also showed that miR-29a expression is higher in the limbs of individuals with DM compared to non-DM [26].

In the current study, we investigated whether the expression of ADAM12 and miR-29a regulation is impaired in the ischemic GA muscle of mice that had been fed a high fat diet (HFD) in a model of type 2 DM. Our laboratory and others have shown that the HFD mice show impaired perfusion recovery and impaired angiogenesis following experimental PAD compared to control mice on a normal chow diet (non-diabetic) [27,28]. How the underlying metabolic derangements in type 2 DM contribute to impaired adaptive mechanisms such as post-ischemic perfusion recovery and skeletal muscle regeneration is poorly understood. Elucidation of these mechanisms may lay the foundation for future work to discover new treatment paradigms. 

In this study, having discovered that ADAM12 upregulation and miR-29a downregulation is also impaired in ischemic GA muscle of mice with DM2 (Figure 1A,B), we then hypothesized that the overexpression of ADAM12 or inhibiting its regulatory miR-29a will improve perfusion recovery and skeletal muscle function in experimental PAD. We also compared the therapeutic effectiveness of both approaches to each other and explored possible cellular and molecular mechanisms involved in the improved outcomes. 

Our results show that ectopic expression of the ADAM12 gene or inhibiting its regulatory microRNA (miR-29a) modify the severity of experimental PAD in mice with DM2. Both interventions resulted in improved perfusion recovery, increased angiogenesis, improved skeletal muscle force production, and reduced skeletal muscle injury in the treated DM2 mice (Figure 4A–C). However, miR-29a inhibition improved perfusion recovery and muscle force production to a greater extent than ADAM12 gene transfer. One possible explanation for this is the observation that miR-29a inhibition leads to greater ADAM12 mRNA and protein expression than ADAM12 overexpression with cDNA (Figure 2). Nevertheless, it has been shown that a single miRNA can regulate multiple genes, often genes that function within the same regulatory pathway [38]. This raises the possibility that the better outcomes observed with the inhibition of miR-29a may be the result of the effects of miR-29a on genes other than ADAM12 involved in post-ischemic muscle recovery. Interestingly, prior studies have shown elevated levels of miR-29 in aged skeletal muscles [39]. Moreover, the overexpression of miR-29 in muscle progenitor cells resulted in impaired proliferation [39]. There is also evidence that in mouse and human myoblasts, Fibroblast growth factor 2 (FGF2)-induced myoblast proliferation is mediated by miR-29a [40], whereas miR29a overexpression regulated muscle atrophy [41]. Taken together with the findings from our studies, this suggests that miR-29a inhibition may be impacting genes involved in the regulation of skeletal muscle post-ischemic adaptation and may contribute to the improvements seen with miR-inhibition in our studies. Although beyond the scope of this study, identifying the other gene targets of miR29a in post-ischemic mouse hind limbs may provide further insight.

Our evaluation of possible cellular mechanisms involved showed both therapeutic interventions were also associated with increased recruitment of endothelial progenitor cells into the ischemic GA muscle of type 2 DM mice (Figure 5B). In addition, our studies and that of others have implicated the receptor tyrosine kinase Tie2 and its downstream signaling as mechanisms involved in post-ischemic adaptation and ADAM12 function [26,34]. Consistent with those findings, we find ADAM12 overexpression and inhibition of miR-29a increased cleaved Tie2 and phosphorylated AKT in ischemic GA muscle of type 2 DM.

Taken together, our data showed impaired ADAM12 upregulation and miR-29a downregulation in ischemic GA muscle of mice with type 2 DM contributes to poorer experimental PAD outcomes in type 2 DM and this can be improved through gene therapy with overexpression of ADAM12 and inhibition of miR-29a. Moreover, the improvements in experimental PAD outcomes following these interventions are likely through increased recruitment of endothelia progenitor cells and increased expression of cleaved Tie2 and AKT phosphorylation.

## 4. Methods

### 4.1. Animals

All experiments were performed under protocols approved by the University of Iowa Institutional Animal Care and Use Committee. Male C57BL/6 mice between 26 and 28 weeks age were obtained from Jackson Laboratory (Bar Harbor, MA). Mice were either fed a normal chow diet (catalog number 000664) or a high-fat diet, HFD, from week 6 after birth (catalog number 380056). Upon arrival, the C57BL/6 mice in the lean control group were fed a standard pelleted rodent chow (Envigo, catalog number 7913) and the high-fat-diet (HFD) mice were given unrestricted access to a pelleted high-fat diet (HFD; 60 kcal%; Teklad Custom Diet, TD.06414). Glucose tolerance was assessed by intraperitoneal GTT (IPGTT) in unanesthetized mice. Briefly, 1 mg/g mouse of glucose was administered intraperitoneally, and glucose was measured at 0, 15 min, 30 min, 1 and 2 h. The area under the curve (AUC) produced was adjusted by the weight of the animal and compared between the groups, and mice within 640 and 1375 AUC-weight adjusted were selected as having impaired glucose tolerance and used for randomization in the experiments (Appendix A). Experimental HFD groups were randomized into 3 treatment groups that received saline (HFD Control), miR-29a inhibitor (HFD + miR-29aINH), or ADAM12 expression plasmid (HFD + ADAM12).

### 4.2. Hind Limb Ischemia Surgery and Perfusion Imaging

Hind limb ischemia surgery was performed as described previously [11,25,26]. Briefly, mice were anesthetized by intraperitoneal injection of a mix of xylazine (5 mg/kg) and ketamine (100 mg/kg) followed by ligation and excision of the femoral artery of the left hind limb. Mice were kept warm (37 °C) throughout the procedure. The right hind limb served as the non-ischemic control, and peripheral blood flow was measured by Laser Speckle Contrast Imaging using PeriCam PSI (Perimed, United States). Images were obtained immediately following surgery (day 0) as well as at day 3, day 7, day 14, and day 21 after the surgery. The relative changes of blood flow in the GA muscle were expressed as the ratio of the operated to the contra-lateral hind limb blood flow using the manufacturer’s software.

### 4.3. In Vivo miR-29a Inhibition and ADAM12 over-Expression

The miR-29a inhibitor and ADAM12 cDNA plasmid were purchased from ThermoFisher, Cat # 4464084 and OriGene Technologies, Cat #: MR225285, respectively. The miR-29a inhibitor and ADAM12 were mixed with specially formulated lipid nanoparticles that would disperse and release their nucleic acid load once the target site is treated with ultrasound [26]. The lipid nanoparticle/nucleic acid complex was delivered 3 days and 1 day before surgery by intra-ocular injection followed by an ultrasound-triggered release in the hind limb as previously described [26].

### 4.4. Muscle Contractile Function

Maximal isometric torque of ankle dorsiflexors, of which the tibialis anterior (TA) muscle is the major agonist, was assessed in control and ischemic limbs using the 1300A 3-in-1 Whole Animal System (Aurora Scientific, Aurora, ON, Canada) as previously described [42]. Briefly, the animals were anesthetized by 3% isoflurane via a nose cone throughout the measurement process. The tibia was stabilized at the knee, and the foot was immobilized with an adhesive tape to a footplate that is attached to a force transducer. Resting tension and muscle length were iteratively adjusted for each muscle to obtain the optimal twitch contraction force. The ankle dorsiflexors were stimulated by subcutaneous electrodes via the fibular nerve. The proper electrode position was determined by a series of isometric twitches. After a 5-min equilibration period, maximal isometric tension was evaluated with stimulations of 150 Hz for 300 ms. Data were collected and analyzed to determine muscle torque using Dynamic Muscle Analysis software (ASI 611A v.5.321; Aurora Scientific).

### 4.5. RNA, Quantitative PCR

Total RNA was isolated using miRNeasy Mini Kit (Qiagen, Cat #2172004), following the manufacturer’s instructions. RNA samples were quantified using a NanoDrop 1000 (Thermo Scientific, Wilmington, DE, USA).

ADAM12 transcript expression was detected by real-time quantitative RT-PCR using TaqMan assays (Applied Biosystems, Foster City, CA), as described previously [11,25,26]. Equal amounts of RNA samples were reverse transcribed using high-capacity RNA to a cDNA synthesis kit (Applied Biosystems, Cat #: 4388950) and 10 ng of cDNA was used for QPCR assay as previously described [11,25,26]. The relative change in expression was determined by the ∆∆Ct method using *Gapdh* (ThermoFisher, Cat: Mm4351370_g1) expression as the reference gene.

To quantify miRNA, RT-PCR was performed using the TaqMan MicroRNA reverse Transcriptase kit (ThermoFisher, Cat: 4366597) according to the manufacturer’s recommendations and as previously described [11,25,26]. Relative change in miR-29a expression was determined by the ∆∆Ct method [11,25,26], using the expression of miR-10a as the reference miRNA. The final fold expression changes were calculated using the equation 2^−ΔΔCt^.

### 4.6. Western Blot Analyses

The total protein from gastrocnemius (GA) or tibialis anterior (TA) muscles was isolated using a radioimmunoprecipitation assay buffer (RIPA buffer-Thermofisher Scientific) according to the manufacturer’s protocol. Homogenates were separated on 4–12% Bis-Tris Gels (Invitrogen), and transferred to nitrocellulose membranes (BioRad,) [11,25,26]. ADAM12 protein bands were detected by the anti-ADAM12 antibody (1:1000 dilution, cat number MBS9128704, MyBio Source), using a secondary antibody conjugated to an infrared dye (1:10,000 dilution IRDye 800CW; LI-COR Biosciences, Lincoln, NE, USA). Signals were recorded using an iBright imager (Thermo Fisher, Model 1500). Bands from Western blots were quantified using the Image Studio Lite software (LiCor). The intensity of protein band signals in different lanes was normalized to the total protein content in each lane on Ponceau S-stained membranes. We have found that the expression of most housekeeping genes is altered in ischemia (unpublished personal observation); moreover, studies have shown total protein staining to be more accurate than the use of a housekeeping protein as a loading control [43].

### 4.7. Immunohistochemistry

For immunohistochemistry, tissue samples were fixed in 2% Paraformaldehyde (PFA) as described previously [11,25,26]. Briefly, 10-µm-thick sections of ischemic and nonischemic hindlimb muscles [gastrocnemius (GA)] were subjected to H&E staining for muscle morphology analysis. Capillaries were identified using a rat anti-mouse CD31 antibody (Novus Biochemicals, Cat # NB100-64796) at 1:1000 dilution. All antibodies were diluted in phosphate buffered saline (PBS) + 1% bovine serum albumin (BSA). Controls with only secondary antibodies were included for all staining described in this manuscript. Slides were scanned (at 20× magnification). All stained cells and muscle fibers in the entire sections were counted, and the capillary density was expressed as capillaries/fibers as illustrated in Appendix A. The percentage of tissue injury was expressed as the total number of centrally located muscle fibers nuclei to the total number of muscle fibers [29].

### 4.8. Flow Cytometry

For flow cytometry analysis, skeletal muscles including the TA and GA muscle from each limb were harvested. The skeletal muscle from each limb was pooled and processed. The muscle tissue was minced into small pieces and digested in 0.5% collagenase type II (Worthington Biochemical, Lakewood, NJ, USA) for 45 min at 37 °C and another 45 min at 37 °C shaking (200 RPM). Digested tissue was passed through 70-μm nylon filters (Becton Dickinson, Franklin Lakes, NJ, USA), and cells in the flow-through were collected by centrifugation (1500× *g* for 15 min). Cells were resuspended in 10mL FACS buffer (PBS, 0.001% NaN3 and 0.04% Fetal Bovine Serum). Cell numbers were determined using a hemocytometer. For each sample, 2 × 10^6^ cells were incubated with Fc receptor blocking (1:400 dilution; BD Biosciences, Cat: BDB553142) for 15 min followed by incubation with PerCP-conjugated anti-mouse CD45 (1:400 dilution; BD Biosciences, Cat: BDB550994), APC-conjugated anti-mouse CD31 (1:400 dilution; BD Biosciences, Cat: BDB551262), and FITC-conjugated CD34 (1:400 dilution; Invitrogen, Cat: 11034182) in dark for 45 min on ice. Cells were then washed 3 times with the FACS buffer and resuspended in 1 mL of the FACS buffer.

Flow cytometry was performed on a Cytek Aurora (Cytek Bioscience) flow cytometer. Cells were gated for single cells by forward- and side-scattered light, and live and dead cells were differentiated by Hoechst 33342 dye (Invitrogen, catalog number: H3570). Subsequently CD45^−^ CD31^+^ CD34^+^ cells were selected as muscle-infiltrating endothelial progenitor cells (EC). Positive events and gates were determined by comparing fluorophore signal intensities between the unstained control and each antibody/fluorophore control.

### 4.9. Statistical Analysis

All measurements are expressed as means ± SEM. Statistical comparisons between two groups at a specific time point were performed using Student’s *t*-test. The comparison of more than two groups was performed with one-way ANOVA. A *p* < 0.05 was considered statistically significant.

### 4.10. GRANTS

The authors disclosed receipt of the following financial support for the research, authorship, and/or publication of this article: National Heart, Lung, and Blood Institute (R01 HL130399 to AO Dokun) and National Institutes of Health (R56AG063820 to V A. Lira).

## Figures and Tables

**Figure 1 ijms-23-00429-f001:**
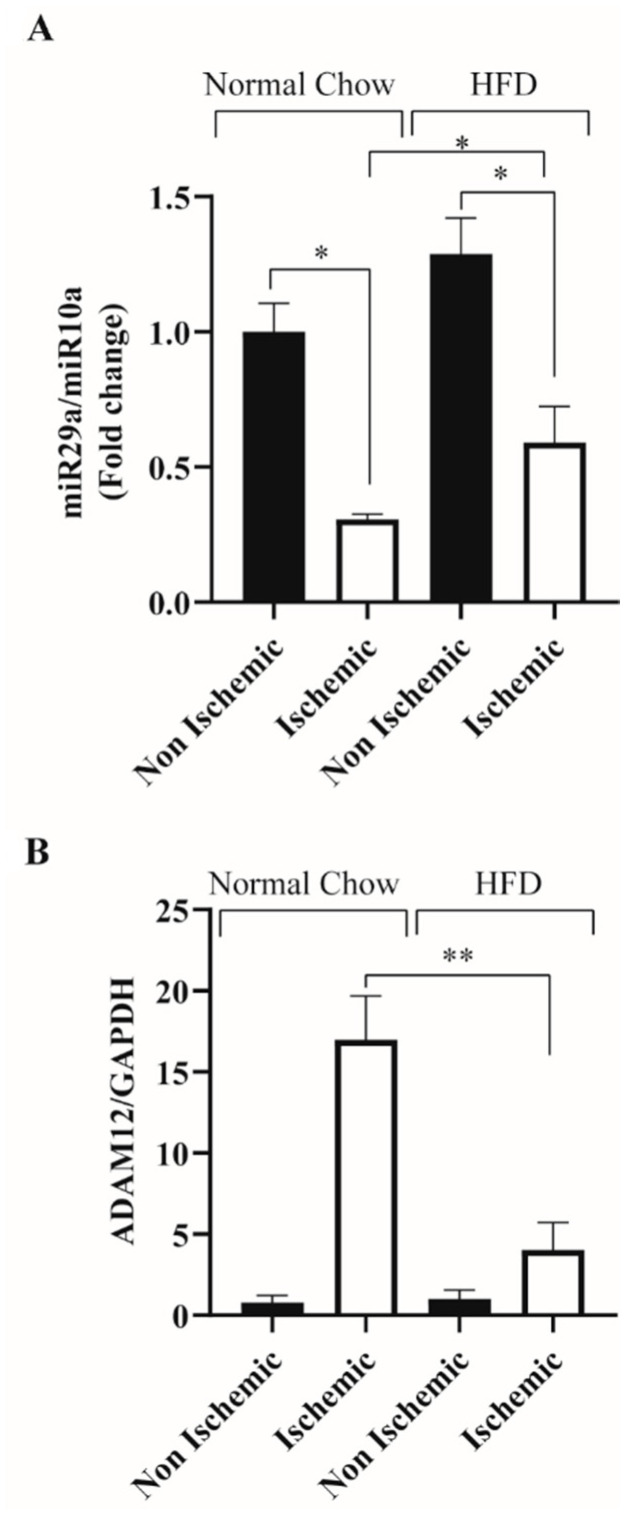
HFD feeding impairs miR-29a downregulation and ischemia induced ADAM12 upregulation: (**A**): 3 days following induction of experimental PAD, there is higher miR-29a expressed in the ischemic GA muscle of mice fed HFD compared to the ischemic hindlimbs of mice fed normal chow (* *p* < 0.05, ** *p* < 0.005, Normal Chow ischemic: *n* = 7, Normal Chow non-ischemic: *n*= 7, HFD ischemic: *n* = 4 and HFD non-ischemic: *n* = 4). (**B**): Ischemic GA muscle of HFD mice show impaired ADAM12 upregulation compared to normal-chow-fed mice (* *p* < 0.05, ** *p* < 0.005, Normal Chow ischemic: *n* = 7, Normal Chow non-ischemic: *n* = 7, HFD ischemic: *n* = 4 and HFD non-ischemic: *n* = 4).

**Figure 2 ijms-23-00429-f002:**
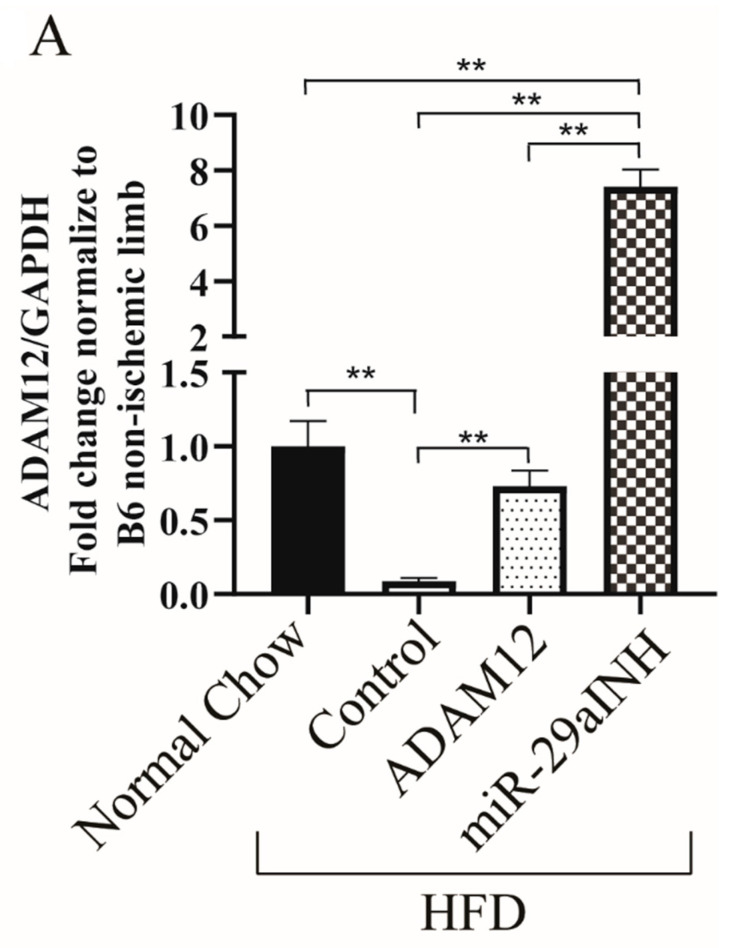
ADAM12 gene transfer and miR-29a inhibition in ischemic GA muscle of HFD mice increased ADAM12 mRNA and protein expression. (**A**): Increased ADAM12 mRNA expression in ischemic GA muscle of HFD mice following ADAM12 gene transfer and miR-29a inhibitor treatment (** *p* < 0.005, Normal Chow: *n* = 5, HFD Control: *n* = 5, HFD + ADAM12: *n* = 4 and HFD + miR29aINH: *n* = 4). (**B**): Increased ADAM12 protein expression in ischemic hindlimbs of HFD mice following ADAM12 gene transfer and miR-29a inhibitor. (**C**): Quantification of ADAM12 expression relative to total protein by ponceaus staining is shown (* *p* < 0.05, ** *p* < 0.005 and Normal Chow: *n* = 5, HFD Control: *n* = 5, HFD + ADAM12: *n* = 5 and HFD + miR29aINH: *n* = 5).

**Figure 3 ijms-23-00429-f003:**
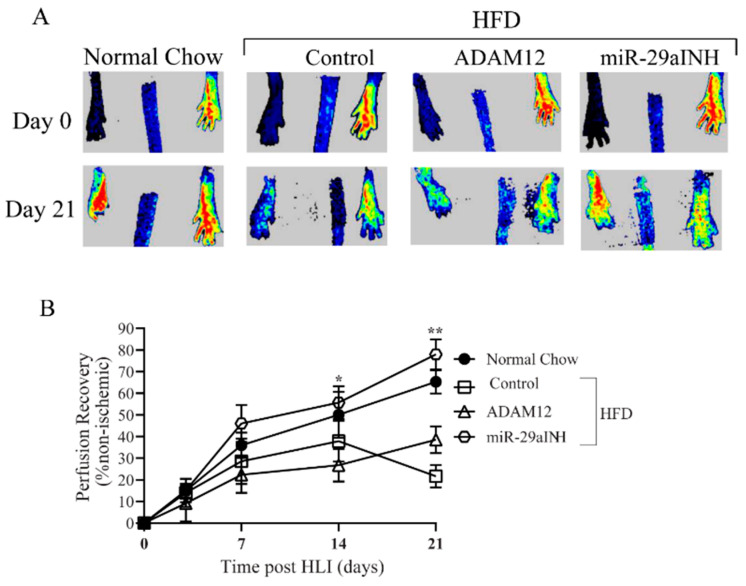
ADAM12 gene transfer and inhibition of miR-29a improved perfusion recovery and capillary density in HFD mice. (**A**): Representative laser speckle contrast imaging (LSCI) showing increased perfusion in post-ischemic limb of HFD mice treated with ADAM12 gene transfer or miR-29a inhibitor. (**B**): Quantification of perfusion recovery post HLI. *y*-Axis shows extent of perfusion in the ischemic limb relative to the non-ischemic limb; *x*-axis shows days following experimental PAD (day-21 percent perfusion recovery * *p* < 0.05, ** *p* < 0.01 and Normal Chow: *n* = 6, HFD Control: *n*= 6, HFD + ADAM12: *n* = 7 and HFD + miR29aINH: *n*= 8). (**C**): GA muscle sections from day 21 post-ischemic GA muscle from normal-chow-fed mice, HFD mice, HFD + ADAM12 and HFD + miR-29a inhibitor treated animals were subjected to immunohistostaining for CD31 to identify capillaries. Representative images of stained sections are shown (scale bar = 20×). (**D**). Quantification of CD31^+^ cells per muscle fiber (* *p* < 0.05 and Normal Chow: *n* = 4, HFD Control: *n*= 4, HFD + ADAM12: *n* = 4 and HFD + miR29aINH: *n*= 4).

**Figure 4 ijms-23-00429-f004:**
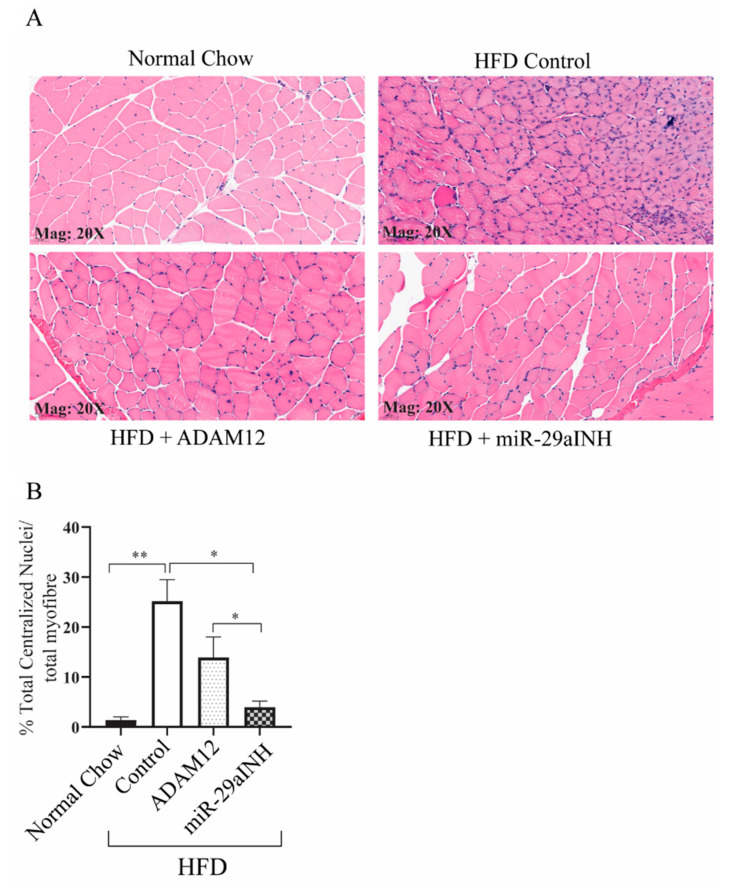
ADAM12 augmentation and miR-29a inhibition reduced the extent of skeletal muscle injury and increased skeletal muscle function in post-ischemic GA muscle of HFD mice. Sections from normal chow, HFD, HFD + ADAM12, and HFD + miR-29aINH treated animals were subjected to immunohistostaining using H&E (for morphology analysis). (**A**) Representative images of H&E-stained sections of day 21 ischemic GA muscle showing centrally located nuclei in muscle fibers. (**B**) Quantification of skeletal muscle injury as determined by the percentage of total number of muscle fibers with centrally located nuclei (* *p* < 0.05, ** *p* < 0.005, *n*= 4/group scale bar = 20×). (**C**) Muscle function analysis as assessed by maximum muscle contraction on 21 days post-ischemic mouse GA muscle (* *p* < 0.05, ** *p* <0.005 and Normal Chow: *n* = 6, HFD Control: *n* = 6, HFD + ADAM12: *n* = 7 and HFD + miR29aINH: *n* = 8).

**Figure 5 ijms-23-00429-f005:**
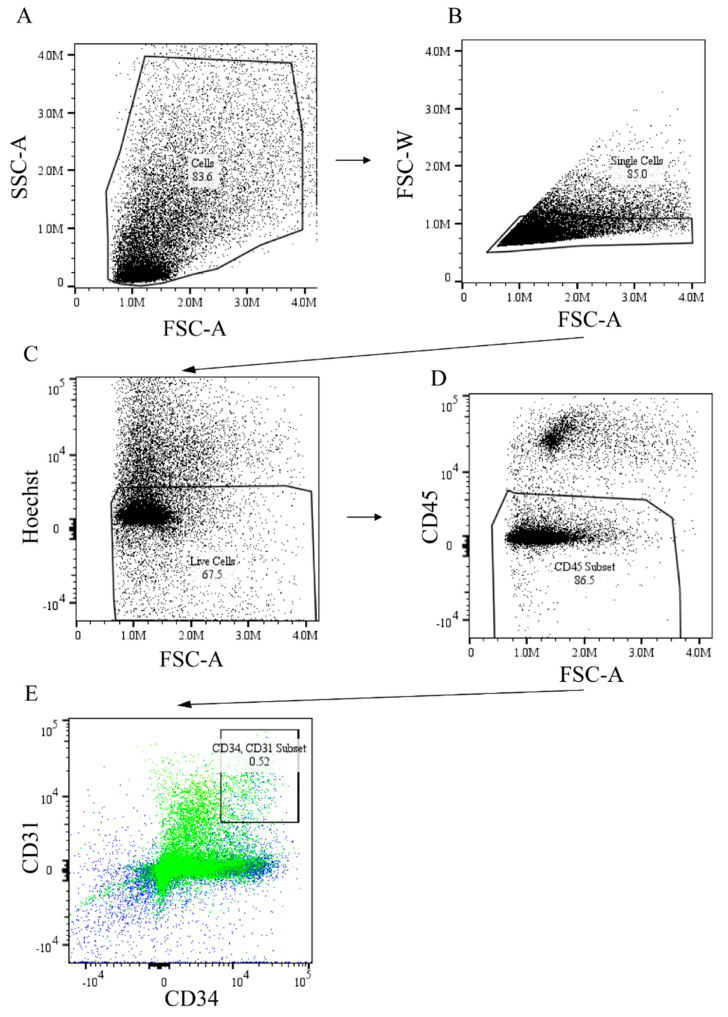
ADAM12 gene transfer and miR-29a inhibition rescued impaired endothelial progenitor cells recruitment in post-ischemic GA muscle of HFD mice. Cells were isolated from day 7 post-ischemic mouse GA muscle from normal chow, HFD, HFD + ADAM12, and HFD + miR-29aINH treated animals and analyzed with flow cytometry. Endothelial progenitor cells were defined as CD45^−^ CD31^+^ CD34^+^ cells. (**A**–**E**): Representative gating of flowcytometric analysis of the EPCs. F: Quantification of the percentage of EPCs in ischemic hind limb of the groups of animals studied (* *p* < 0.05 and Normal Chow: *n* = 4, HFD Control: *n* = 5, HFD + ADAM12: *n*= 4 and HFD + miR29aINH: *n* = 5).

**Figure 6 ijms-23-00429-f006:**
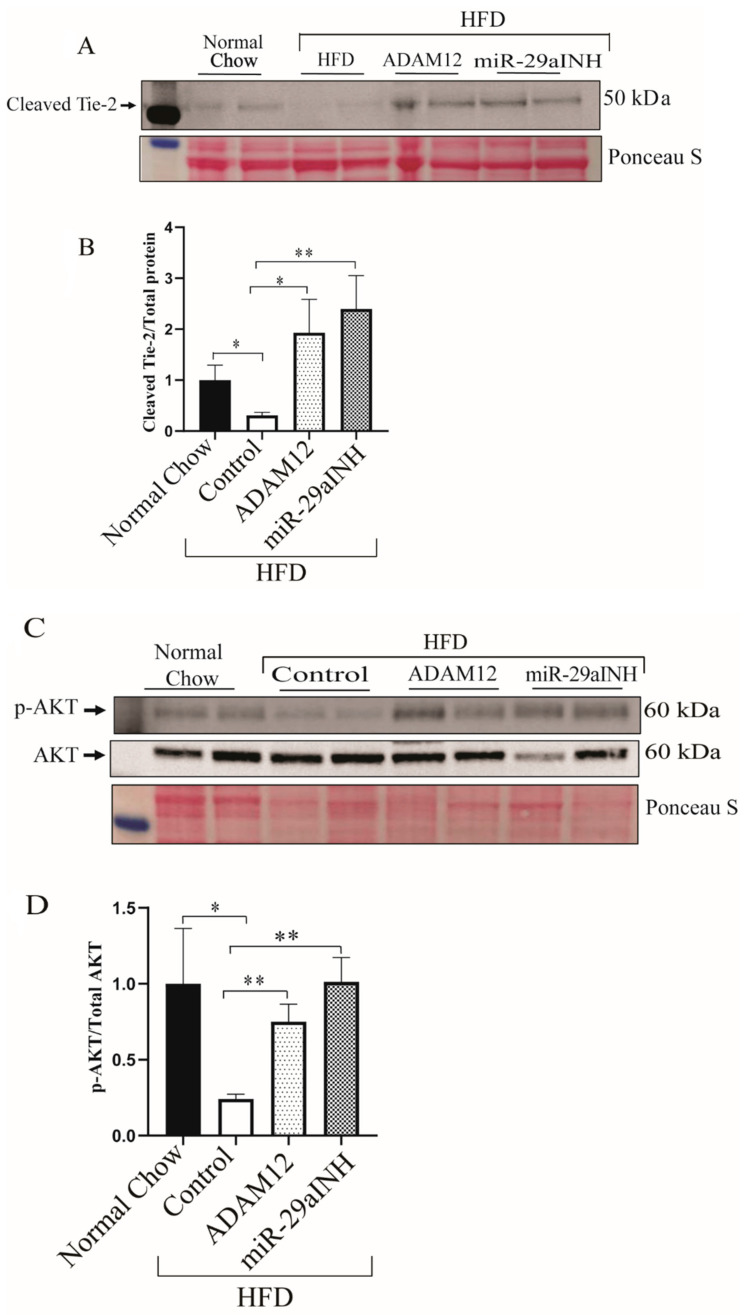
miR-29a inhibition and ADAM12 augmentation modulates Tie2 cleavage through increased Akt phosphorylation (p-AKT) in vivo: (**A**): Tie2 cleavage western showing higher expression of cleaved Tie2 in GA lysates from mice with type 2 diabetes hindlimbs that received miR-29aINH (HFD + miR-29aINH) and ADAM cDNA (HFD + ADAM12) compared with those that received control plasmid (HFD Control), 3 days post-HLI. (**B**): Quantification of Western blot bands (*n* = 6, * *p* < 0.05, ** *p* < 0.005 and Normal Chow: *n* = 6, HFD Control: *n* = 6, HFD + ADAM12: *n* = 7 and HFD + miR29aINH: *n* = 8). (**C**): Akt and p-Akt Western showing higher expression in muscle lysates from mice with type 2 diabetes hindlimbs that received miR-29aINH (HFD + miR-29aINH) and ADAM cDNA (HFD + ADAM12) compared with those that received control plasmid (HFD Control), 3 days post-HLI. (**D**): Quantification of Western blot bands (* *p* < 0.05, ** *p* < 0.005 and Normal Chow: *n* = 6, HFD Control: *n* = 6, HFD + ADAM12: *n* = 7 and HFD + miR29aINH: *n* = 8).

## Data Availability

Not applicable.

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
