# Peer review of "Modulation of miR-29a and ADAM12 Reduces Post-Ischemic Skeletal Muscle Injury and Improves Perfusion Recovery and Skeletal Muscle Function in a Mouse Model of Type 2 Diabetes and Peripheral Artery Disease"

_ijms, 2021, doi:10.3390/ijms23010429_

Round 1

Reviewer 1 Report

Authors have a thorough understanding of the miR-29a, ADAM12 and Tie2 pathway because of their prior work in DM1. This study is focused on the DM2 scenario using HFD.

At high level the average reader will have a hard time understanding the mechanism of how this pathway works and why the progression of experiments followed the way it did. The progression of experimental strategies need to be better connected in the discussion sections and introduction because the logical scientific approach and analysis is sometimes missing.

For example

  1. the significance of this work in PAD and therapy is irrelevant in the beginning of the discussion and very repetitive to what was already mentioned in the introduction. Valuable real estate in the discussion section (line 262-284) is wasted in convincing the reader why this work is relevant.
  2. Similarly, In sections 286 to 302 a lot of time is devoted to explain strain differences and little on the observed results and its significance.
  3. Towards the end of the discussion claims are made of how this approach of ADAM12 overexpression and miR-29a inhibition will lead to improved outcomes. This is pre-emptive without knowing the preclinical pathway reasonably well especially in situations like this where there are feedback loops that may have unintended side effects or consequences. While the results are promising these discussions needs to be carefully explained with evidence and progress in the field detailing the promise.
  4. The authors need to rewrite some of the sections to ensure these points are made clear.

Author Response

Response to Reviewers’ Comments

Reviewer-1.

The significance of this work in PAD and therapy is irrelevant in the beginning of the discussion and very repetitive to what was already mentioned in the introduction. Valuable real estate in the discussion section (line 262-284) is wasted in convincing the reader why this work is relevant.

Response

We thank the reviewer for this insightful comment we have now removed the repetitive section from the discussion.

Similarly, in sections 286 to 302 a lot of time is devoted to explain strain differences and little on the observed results and its significance.

Response

We have removed the aspects of the discussion that focused on strain differences.

Towards the end of the discussion claims are made of how this approach of ADAM12 overexpression and miR-29a inhibition will lead to improved outcomes. This is pre-emptive without knowing the preclinical pathway reasonably well especially in situations like this where there are feedback loops that may have unintended side effects or consequences. While the results are promising these discussions needs to be carefully explained with evidence and progress in the field detailing the promise.

The authors need to rewrite some of the sections to ensure these points are made clear.

Response

We have removed the aspects of the discussion that focused on strain differences. Additionally, we have added additional discussion about the implications of the study in the context of some prior relevant studies (315-327). Lastly we agree that we may have overstated the implications of the study. We have tempered our statements, limiting it to our actual findings (336-341).

Reviewer 2 Report

In this manuscript Lamin et al investigate in the hindlimb the regulation of ADAM12 and miR-29a in post-ischemic condition using a mouse model of type 2 diabetes. In addition, the authors also explore whether the modulation of ADAM12 and miR-29a can improve perfusion recovery in this model. The results obtained are interesting but I have some criticisms.

Major criticisms:

- In all the figures, the legend indicates hindlimbs. Be more precise! Which muscle is it?

- In your experiments, what is exactly the number of animals per groups? Depending on the experiment, the number indicated in the legend goes from 3 to 9. 

If each group contains 9 mice, why not do all the experiments with n=9? Otherwise it suggests that you have chosen animals or results and that the results are biased! n=3 or 4 is really too few! This is a really important point to clarify.

- For the quantification of CD31+ cells and central nuclei from the histological sections of muscle, which method did you use (nanozoomer, software...)?  Which muscle? How many fibers did you really count?

- In figure 4 you have studied the muscle strength. This force depends on the metabolic and contractile characteristics of the muscle fibers and the muscle mass. What is the impact of ADAM12 and miR-29a on muscle weight (quadriceps, gastrocnemius and tibialis).

- Line 319-321 you wrote: “This raises the possibility that inhibition of miR-29a may result in better outcomes than overexpression of ADAM12 by impacting other, yet to be determined, genes involved in post ischemic adaptation.”. This is obvious. By consulting databases on miRNAs, we see that miR-29a potentially regulates several hundred genes. The role of miR-29a in muscle has been described for some years now, especially during muscle regeneration, atrophy processes or during aging.

In conclusion, the manuscript is interesting but it leaves the reader with a feeling of approximation and a doubt on the real results insofar as only certain mice seem to be taken into account according to the analyses made.

Author Response

Response to Reviewers’ Comments

Reviewer-2

In all the figures, the legend indicates hindlimbs. Be more precise! Which muscle is it?

Response

We thank the reviewer for this insightful comment. We have modified the manuscript to indicate that the gastrocnemius muscle or GA was used throughout the study (red highlight, figure legend to all figures).

In your experiments, what is exactly the number of animals per groups? Depending on the experiment, the number indicated in the legend goes from 3 to 9.  If each group contains 9 mice, why not do all the experiments with n=9? Otherwise it suggests that you have chosen animals or results and that the results are biased! n=3 or 4 is really too few! This is a really important point to clarify.

Response

We thank the reviewer for this comment, and we provide the following modifications and clarifications. Although we start off with the same number of animals in each experimental group, following the hind limb ischemia surgery sometimes mice are lost in some of the experimental groups resulting in unequal number of animals in each group. We have now clarified the number of mice used in each experimental group within the legend of each figure (highlighted in red).  We have increased the number of animals to ensure there are no experiments in which we used less than 4 mice per group. It is clear from our results that we can detect differences between experimental groups even in those experiments in which there are 4 mice in an experimental group. This suggests despite the use of 4 mice in some of the studies we are powered to detect a difference between the experimental groups. Moreover, in most of the experiment the number of mice per group is >4.

For the quantification of CD31+ cells and central nuclei from the histological sections of muscle, which method did you use (nanozoomer, software...)?  Which muscle? How many fibers did you really count?.

Response

We thank the reviewer for this question and for the opportunity to provide clarification. CD31+ cells and muscle fibers with centrally localized nuclei were manually counted by an observer who is blinded to the experimental design and experimental groups. The capillaries and muscle fibers were counted in the gastrocnemius muscle sections. All fibers within a section were counted and all CD31+ cells within the section were counted. Although this approach is time consuming and tedious, we found it to be more reliable.   The number of capillaries/muscle fiber and muscle fibers with centrally placed nuclei is presented in the figures (3c-d and 4a-b).

In figure 4 you have studied the muscle strength. This force depends on the metabolic and contractile characteristics of the muscle fibers and the muscle mass. What is the impact of ADAM12 and miR-29a on muscle weight (quadriceps, gastrocnemius and tibialis)

Response

We thank the reviewer for this question. The muscle mass was not measured in these experiments and therefore we do not know the impact of ADAM12 and miR29a inhibition on muscle weight. We agree that this may be provide additional useful information and consider including this in the analysis in future studies.

Line 319-321 you wrote: “This raises the possibility that inhibition of miR-29a may result in better outcomes than overexpression of ADAM12 by impacting other, yet to be determined, genes involved in post ischemic adaptation.”. This is obvious. By consulting databases on miRNAs, we see that miR-29a potentially regulates several hundred genes. The role of miR-29a in muscle has been described for some years now, especially during muscle regeneration, atrophy processes or during aging.

Response

We thank the review for this insightful comment. We agree with the reviewer that miR29a has the potential to regulate hundreds of genes and its role on muscle have been described in the context of atrophy, ageing and growth factor induced muscle regeneration.  We have now edited our discussion to include some of these studies (Hu et.al, Aging. 2014 Mar; 6(3): 160–175, Galimov et al. Stem Cells. 2016 Mar;34(3):768-80. and Wang et.al Hum Gene Ther. 2020 Mar;31(5-6):367-375.) and discuss our findings with these prior studies in mind (lines 315-327). 

Reviewer 3 Report

The Authors present a paper entitled “In a Mouse Model of Type 2 Diabetes and Peripheral Artery 2 Disease, Modulation of miR-29a and ADAM12 Reduced Post - 3 Ischemic Skeletal Muscle Injury, Improved Perfusion Recovery 4 and Skeletal Muscle function”. They have demonstrated that the expression of miR-29a and ADAM12 resulted impaired in ischemic hind limbs of DM2 mouse model and that their modulation may pave the way to improve clinical outcomes in peripheral artery disease. The problem is well presented, and the background sufficiently described in the Introduction. The statistical analysis and the Materials and Methods description is appropriate. The authors suggested that ADAM12 gene transfer and miR-29a inhibition improved outcomes in ischemic hind limbs of DM2 mice, increasing skeletal muscle function and reducing skeletal muscle injury. About this, it was demonstrated that therapeutic interventions that improve perfusion simultaneously with skeletal muscle abnormalities may have the greatest effects on walking impairment in people with PAD. In this direction, it would be interesting to investigated if overexpression of ADAM12 or inhibiting its regulatory miR-29a may improve walking performance in DM2 mouse model using treadmill exercise. Perhaps we will see this in a future report.

Author Response

Response to Reviewers’ Comments

Reviewer-3

The Authors present a paper entitled “In a Mouse Model of Type 2 Diabetes and Peripheral Artery 2 Disease, Modulation of miR-29a and ADAM12 Reduced Post - 3 Ischemic Skeletal Muscle Injury, Improved Perfusion Recovery 4 and Skeletal Muscle function”. They have demonstrated that the expression of miR-29a and ADAM12 resulted impaired in ischemic hind limbs of DM2 mouse model and that their modulation may pave the way to improve clinical outcomes in peripheral artery disease. The problem is well presented, and the background sufficiently described in the Introduction. The statistical analysis and the Materials and Methods description is appropriate. The authors suggested that ADAM12 gene transfer and miR-29a inhibition improved outcomes in ischemic hind limbs of DM2 mice, increasing skeletal muscle function and reducing skeletal muscle injury. About this, it was demonstrated that therapeutic interventions that improve perfusion simultaneously with skeletal muscle abnormalities may have the greatest effects on walking impairment in people with PAD. In this direction, it would be interesting to investigated if overexpression of ADAM12 or inhibiting its regulatory miR-29a may improve walking performance in DM2 mouse model using treadmill exercise. Perhaps we will see this in a future report.

Response:

We thank the reviewer for this favourable review of our manuscript and in future studies we hope to include the impact of overexpression of ADAM12 and inhibition of miR-29a on walking performance in DM2 mice using the treadmill exercise as suggested.

Round 2

Reviewer 2 Report

I am not convinced by your explanation of the number of animals used. I have doubts about the veracity of your statements. Especially when you say that you have counted manually CD31+ cells and muscle fibers with centrally localized nuclei on gastrocnemius sections?

According to your indications, you studied 16 mice for CD31+ labeling and as many to determine the central nuclei, that represents 32 historical sections in all if I am not mistaken.

Do you know how many fibers there are in a gastrocnemius? There are about 12000. Do you want me to believe that you have manually counted about 360,000 fibers? This is not possible. For this type of analysis counting software is used, otherwise you have only counted a few fields.

What is the truth?

Author Response

I am not convinced by your explanation of the number of animals used. I have doubts about the veracity of your statements. Especially when you say that you have counted manually CD31+ cells and muscle fibers with centrally localized nuclei on gastrocnemius sections?

According to your indications, you studied 16 mice for CD31+ labeling and as many to determine the central nuclei, that represents 32 historical sections in all if I am not mistaken.

Do you know how many fibers there are in a gastrocnemius? There are about 12000. Do you want me to believe that you have manually counted about 360,000 fibers? This is not possible. For this type of analysis counting software is used, otherwise you have only counted a few fields.

What is the truth?

We again thank the reviewer for sharing his perspective and we understand why he/she might doubt that we manually counted 32 sections with thousands of muscle fibers. We also recognize that this is very tedious work, but this is in fact what we did.  Of the 32 sections that were used, 16 were stained with H&E and used for fiber number and centralized nuclei analysis while the remaining 16adjacent sections were used for CD31 staining.  We manually counted all GA fibers from the 16 H&E-stained slide and this resulted in a total of 99,623 fibers (please see now added supplementary data table 7a). We also manually counted all CD31+ cells in 16 sections adjacent to the H&E-stained sections resulting in 28,960 cells counted (supplementary data Table 7b).  To calculate the capillaries per fiber we divided the number of capillaries by the number of fibers counted from the adjacent H&E-stained sections. The number of muscles with centrally place nuclei also came from the same 16 H&E-stained sections and this showed a total of 9179 fibers (supplemental data Table 7c).  This manual counting took 2 students several weeks to complete.  The average number of total fibers we obtained from our mouse GA are lower than that suggested by the reviewer.

Our average fiber/GA is about 6220 with a few samples showing total fiber counts >10,000. One possible explanation for this is that we are counting post ischemic GA's. We could not find data on prior studies assessing the total number of fibers in ischemic GA's to compare to our data. It is however well known that muscle fiber necrosis occurs in the ischemic GA and therefore these GA would be expected to have lower number of GA’s. We hope this provides the necessary clarity to and transparency to the reviewer.

Round 3

Reviewer 2 Report

Ok.